# The Impact of Cigarette Smoking on Risk of Rheumatoid Arthritis: A Narrative Review

**DOI:** 10.3390/cells9020475

**Published:** 2020-02-19

**Authors:** Yuki Ishikawa, Chikashi Terao

**Affiliations:** 1Section for Immunobiology, Joslin Diabetes Center, Harvard Medical School, One Joslin Place, Boston, MA 02215, USA; yuki.ishikawa.sw@riken.jp; 2Laboratory for Statistical and Translational Genetics, Center for Integrative Medical Sciences, RIKEN, 1-7-22 Suehiro-cho, Tsurumi-ku, Yokohama, Kanagawa 230-0045, Japan; 3Clinical Research Center, Shizuoka General Hospital, 4 Chome-27-1 Kitaando, Aoi Ward, Shizuoka 420-8527, Japan; 4Department of Applied Genetics, The School of Pharmaceutical Sciences, University of Shizuoka, Shizuoka 422-8526, Japan

**Keywords:** rheumatoid arthritis (RA), anti-citrullinated cyclic peptide/protein antibody (ACPA), rheumatoid factor (RF), etiology, genetics, *HLA-DRB1*, shared epitope allele, environmental risk factors, cigarette smoking, single nucleotide polymorphism (SNP)

## Abstract

Rheumatoid arthritis (RA) is an autoimmune disease characterized by chronic inflammation and subsequent proliferation of synovial tissues, which eventually leads to cartilage and bone destruction without effective treatments. Anti-citrullinated cyclic peptide/protein antibody (ACPA) and rheumatoid factor (RF) are two main characteristic autoantibodies found in RA patients and are associated with unfavorable disease outcomes. Although etiologies and causes of the disease have not been fully clarified yet, it is likely that interactive contributions of genetic and environmental factors play a main role in RA pathology. Previous works have demonstrated several genetic and environmental factors as risks of RA development and/or autoantibody productions. Among these, cigarette smoking and *HLA-DRB1* are the well-established environmental and genetic risks, respectively. In this narrative review, we provide a recent update on genetic contributions to RA and the environmental risks of RA with a special focus on cigarette smoking and its impacts on RA pathology. We also describe gene–environmental interaction in RA pathogenesis with an emphasis on cigarette smoking and *HLA-DRB1*.

## 1. Introduction

Rheumatoid arthritis (RA) is an autoimmune disease characterized by chronic inflammation and subsequent proliferation of synovial tissues, which eventually leads to cartilage and bone destruction without effective treatments [1]. The prevalence of RA is reported to be 0.5–1.0% according to most epidemiologic studies [2].

Anti-citrullinated cyclic peptide/protein antibody (ACPA) and rheumatoid factor (RF) are two main characteristic autoantibodies found in 70–80% of RA patients. It is now well known that not only positivity for, but also high levels of, both of these autoantibodies have associations with joint destruction [3,4,5,6,7] and systemic bone loss even in early phases of the disease course [8,9]. Moreover, about 50–60% of patients have both ACPA and RF, and both of these autoantibodies show an additive effect on the amount and extent of bone erosion and thus disease severity [6,10]. This is why these autoantibody profiles of patients are considered to be one of the disease’s prognostic markers and are included in the 2010 ACR/European League Against Rheumatism (EULAR) classification criteria for RA [11]. In contrast, subjects who meet the classification criteria but are negative for ACPA and RF are considered to be seronegative RA patients. Several different etiopathological aspects have been observed in these patients, and thus, it is reasonable to regard seropositive and seronegative RA as distinctive subtypes. Although seronegative RA has been considered to be a milder form of the seropositive disease [5,6,12], a recent study indicated that treatment response was somewhat slower in seronegative patients, and radiographic progression was similar in seronegative and seropositive patients, suggesting that seronegative RA is not a mild form of the disease and requires intensive therapy similar to seropositive RA [13].

A growing body of evidence has accumulated, and there has been remarkable progress in understanding RA pathogenesis in the last few decades, but a lot of aspects of its precise mechanisms are still unexplained. It is now widely accepted that both environmental and genetic factors contribute to the pathogenesis of RA, and numerous previous works have found that interactive contributions of genetic and environmental factors play a main role in the development of RA [14,15]. However, again, their exact roles in the course of RA development have not been fully clarified yet.

In the following sections, we provide an overview of the genetic risks of RA, followed by the impact of cigarette smoking (CS) on RA pathology and other environmental factors whose effects can be influenced by CS. Subsequently, we describe the effects of these risks on RA pathogenesis more precisely with a special focus on gene-environmental interaction. We used MEDLINE for our literature search of the following terms: rheumatoid arthritis/RA, cigarette smoking, anti-citrullinated cyclic peptide/protein antibody/ACPA, rheumatoid factor/RF, HLA-DRB1, shared epitope allele, environmental risks, genetic risks, and single nucleotide polymorphisms/SNPs. We selected studies where the number of subjects involved was more than 100 in both the case and control for case–control studies and there were more than 100 incidences or RA subjects for cohort studies. For environmental risks other than CS, only those influenced by CS were selected. For genetic risks, only those associated with environmental risks were selected. We also selected meta-analyses, all of which were conducted using enough studies with sufficient numbers of subjects, as described above.

## 2. Genetic Risk Factor of RA

According to the twin study conducted by MacGregor et al., the heritability of RA was estimated to be ~60%, and there was no difference in the overall genetic contribution to RA among variables of sex, age, age at onset, and disease severity [16]. In contrast, genetic variations, mostly represented by single nucleotide polymorphisms (SNPs), also contribute to RA pathogenesis. Genome-wide association studies (GWAS) and GWAS meta-analyses with the use of in silico imputation of SNPs have reported 106 RA risk loci to date [17,18,19,20,21,22,23,24], among which, only ~20% are coding variants while the rest of the ~80% variants in non-coding regions probably regulate gene expression [20].

Of note, recent advances in a genetic analysis of juvenile idiopathic arthritis (JIA) revealed that several genetic risks in adult RA patients, such as *HLA-DRB1*04* or *PTPN22*, also confer a risk or a protection on JIA, especially RF-positive polyarthritis, suggesting shared genetic backgrounds between adult seropositive RA and RF-positive JIA [25,26].

### 2.1. HLA

The major histocompatibility (MHC) region is located at chromosome 6 and contains the human leukocyte antigen (*HLA*) locus. The development of an *HLA* imputation method led to fine mapping of genetic risks of RA within the MHC region. The *HLA* indicates the strongest genetic risk of RA, explaining 30–50% of total genetic risk [27]. Among *HLA* genes, *HLA-DRB1,* one of the class II *HLA* genes, was the first identified RA risk locus [28] and confers the majority of genetic risk of RA [2]. The specific amino acid (AA) sequence at positions 70–74 of HLA-DRβ1 chains is called shared epitope (SE), and SE was reported to explain the association of *HLA-DRB1* with RA susceptibility (SE hypothesis) [28]. As a result of recent advances of imputation for *HLA* sequences, large-scale association studies have revealed that AA positions 11 or 13, 71, and 74 of HLA-DRβ1 are strongly associated with RA in European populations [29,30]. A recent study also revealed a very similar genetic architecture in Asian populations to that in European populations, whereas AA position 57 is unique to Asian populations. The current consensus is that most risk *HLA* variants are shared among different populations at AA levels.

*HLA-DRB1* is also associated with positivity for RF and ACPA [31] and levels of ACPA [32,33] but not for RF [34]. Non-SE alleles, such as *HLA-DRB1*09:01* in Asian populations [32,33], have also been reported to be associated with RA [35] or ACPA levels, and the associations of *HLA-DRB1* with ACPA levels are mainly explained by the 74th AA, alanine [32,33]. Intriguingly, differences in genetic backgrounds between ACPA-positive and ACPA-negative RA were highlighted by a clear difference in signals at the *HLA* region [36], and such differences can also be explained by the same HLA-DRβ1 AA positions but different risk-associated residues [30]. These findings can explain the heterogeneity in clinical manifestations between these RA subtypes and may also imply that other autoimmune-related factors contribute to ACPA-negative RA development [2].

Recent studies have also revealed that AA polymorphisms in other classical *HLA* genes, *HLA-DPB1*, *HLA-B*, and *HLA-A* [29,30,37], and a coding variant in a non-classical *HLA* gene, *HLA-DOA*, which alters the expression levels of several genes, also indicate the risk of ACPA-positive RA [38].

### 2.2. PTPN22

*PTPN22* encodes a protein tyrosine phosphatase that is exclusively expressed in immune cells [39]. The SNP R620W, and the resultant risk allele, lymphoid tyrosine phosphatase (LYP) allele, is the most well-characterized risk variant of RA, as well as multiple autoimmune diseases, including type I diabetes, systemic lupus erythematosus, Hashimoto thyroiditis, Graves’ disease, Addison’s disease, myasthenia gravis, vitiligo, systemic sclerosis, juvenile idiopathic arthritis, and psoriatic arthritis [40]. Intriguingly, the association of the risk variant with RA susceptibility is only found in Caucasians, but not in Asian populations according to the meta-analysis conducted by Nabi et al. [41]. The LYP allele is a gain-of-function variant leading to decreased TCR and BCR signaling, followed by a breakdown of both central and peripheral tolerance [42,43]. Impaired regulatory T cell function [44] and frequency [45] reduced Toll-like receptor 7-induced type I interferon signal [46], and hypercitrullination of peripheral blood mononuclear cells via physical interaction with peptidylarginine deaminase type 4 (PADI4) [47] was also reported in relation to this variant. Each of these effects probably work in a cell-specific manner with respect to the pathogenesis of RA.

### 2.3. PADI4

*PADI4* was identified as the first non-MHC RA risk locus in the Japanese population [48] and was later confirmed in European populations [19]. PADI4 was expressed in myeloid lineage cells and rheumatoid arthritis synovial tissues [48], as well as in *Porphyromonas gingivalis* and *Aggregatibacter actinomycetemcomitans* in gingival tissue [49], and it post-translationally converts peptide-bound arginine residues into citrulline, leading to citrullinated epitope generation, which is recognized by ACPAs [50].

### 2.4. Important Considerations for a Genetic Study

Because allelic variants presenting in more than 1–5% of a given population are identified in GWAS, a number of unusual or rare variants are missed (missing heritability). Missing heritability is hard to analyze with the same statistical methods used in GWAS, and thus, specific statistics for analysis are needed. Moreover, GWAS usually investigate SNPs that are in strong linkage disequilibrium (LD) with other SNPs and serve as proxies for them, and thus, the identified SNPs by GWAS are regarded merely as tags for the yet-to-be-identified causal allele. Next-generation sequencing (NGS) is one of the promising tools for future fine-mapping studies. With the use of NGS, several *HLA*-related genes, including non-classical *HLA* genes, *HLA*-like genes, and pseudo-*HLA* genes, as well as key immune-related genes, can be incorporated into current reference panels, which will enable us to identify disease-related variants with higher resolution.

## 3. Cigarette Smoking as the Most Robust Environmental Risk of RA

Previous studies have shown the contribution of various environmental factors to RA pathogenesis. Inhaled pollutants, especially CS, have been the most extensively studied and a topic of this article (Table 1).

Regardless of the autoantibody status, CS increases the risk of RA development by 26% in those who smoked 1–10 pack-years (a lifelong CS exposure defined by the following formula; pack-years = [number of cigarettes smoked per day/20] × [number of years smoked]) and by 94% in those with more than 20 pack-years according to the meta-analysis conducted by Di Giuseppe et al. [51]. It has been reported that males are more susceptible to CS than females with respect to RA development [52]. CS can even affect treatment response to disease-modifying anti-rheumatic drugs (DMARDs) [53] and thus can be a risk of future joint destruction [54,55,56].

### 3.1. Effects of Intensity and Duration of Cigarette Smoking and Smoking Cessation

Importantly, both CS intensity and duration are directly related to the risk of RA development with prolonged increased risk even after CS cessation [51]. Di Giuseppe et al. conducted a meta-analysis of the association between pack-years and the risk of RA development. Three prospective and seven case–control studies were included in the analyses, and they found that smokers had a higher risk of RA development than never-smokers in a dose-dependent manner up to 20 pack-years, after which the risk did not increase further. Among smokers, RF-positive cases had higher a risk than RF-negative cases [51]. In contrast, Hedström et al. conducted a case–control study with 3655 cases and 5883 controls from a Swedish population [52], in which they found a dose-dependent increase of RA risk in both ACPA-positive and -negative cases, with greater effects in ACPA-positive cases. Interestingly, CS duration had a higher influence on the association of CS and RA than did CS intensity. They also found different effects of CS cessation on ACPA-positive and -negative cases; the association of CS with RA no longer persisted after 20 years of cessation in ACPA-negative cases, while the association persisted with cumulative dose dependency in ACPA-positive cases.

### 3.2. Effects of Passive Smoking

While direct smoking has become an established risk of RA and its disease course, Seror et al. reported that passive smoking during childhood affected susceptibility to RA in the French E3N cohort (98,995 women born between 1925 and 1950) [57,58]. Disease activity was also affected by passive smoking in Korean (191 cases) [59] and Egyptian (100 cases) [60] female RA patients, implicating the importance of avoiding any CS-exposing environment. In contrast, Hedström et al. found that there was no association between passive smoking and RA risk in the EIRA cohort (589 cases and 1764 controls aged 18–70 years) from Sweden, which might be explained by a threshold of smoking intensity below which an association between smoking exposure and RA does not occur [61]. A recent study conducted by Kronzer et al. (1198 cases and 3061 controls) also did not show a clear association between passive smoking and RA risk [62]. However, because those two studies did not measure the effect of childhood exposure to passive smoking and there seemed to be a linear trend between pack-years and ORs in the latter study [62], it may be of interest to know not only whether there is a true association between childhood exposure to passive smoking and RA risk, but also if the association is just a consequence of intensity of smoking exposure or a specific effect of childhood exposure.

### 3.3. Effects of Cigarette Smoking on RA-Related Autoantibody Production

As has been implicated in the studies above and others, the association between CS and RA risk is stronger in seropositive cases than in seronegative cases. It has also been suggested that CS may affect RA-associated autoantibody formation. A study conducted by van Wasemael et al. showed that CS was associated with multiple autoantibody positivity (RF, ACPA, anti-carbamylated protein (CarP) antibody) not only in RA patients of European descent but also in Japanese non-RA subjects [63]. Furthermore, the study also indicated that CS might have a stronger association with RF than with ACPA or anti-CarP antibody. CS may break tolerance to autoantigens in RA, which might be one of the triggers of RA onset in subsets of patients. We also recently reported that CS affected both positivity and levels of ACPA and RF with greater effects on RF using 6239 Asian RA cases, the largest Asian study ever [64]. The study also implicated the dose-dependent effects of CS and the effects of CS cessation on autoantibody levels, the latter of which lasted for up to 20 years both in ACPA and RF cases.

### 3.4. Other Environmetal Risks Augmented by Cigarette Smoking

Several environmental risks have been reported to be influenced by CS. Among them, occupational silica exposure seems to be convincing, while the rest need to be investigated further with well-powered studies for confirmation of the associations.

#### 3.4.1. Occupational Silica Exposure

Occupational exposure to crystalline silica (SiO2), especially in male workers, is a well-known example of environmental risk. Two Swedish studies, a population-based case–control study comprising 577 incident RA cases and 659 randomly selected controls from EIRA [65] and an independent Swedish construction health examination cohort study comprising a total of 240,983 participants [66], reported that risk of RA by silica exposure exceeded that expected from the separate effects of silica and CS among smokers. It was suggested that silica-induced inflammation and fibrosis may be mechanistically separate, because the steps in the development of silicosis, including acute and chronic inflammation and fibrosis, have different molecular and cellular requirements [67]. Autoimmunity would probably start with activation of the innate immune system, leading to proinflammatory cytokine production, pulmonary inflammation, subsequent activation of adaptive immunity, breaking of tolerance, and autoantibody production leading to tissue damage. It also suggests substantial genetic involvement and gene/environment interaction in silica-induced autoimmunity [67].

#### 3.4.2. Alcohol Consumption

Consumption of moderate amount of alcohol has been reported for the beneficial effect on RA development [68,69,70]. It is well accepted that CS and alcohol consumption are common in RA patients, thus both need to be adjusted when the association of each risk with RA is to be investigated, because CS is more prevalent among alcohol drinkers [71]. The study conducted by Källberg et al. comprising the Swedish EIRA cohort (1204 cases and 871 controls) and the Danish CACORA (444 cases and 533 controls) indicated a greater alcohol-related risk reduction for ACPA-positive RA observed in ever-smokers carrying SE alleles compared with never-smokers [72].

#### 3.4.3. Sugar-Sweetened Soda Consumption

Regular consumption of sugar-sweetened soda has been reported to be associated with increased risk of seropositive RA in women, independent of other dietary and lifestyle factors [73]. The study population comprised two cohorts, the Nurses’ Health Study (NHS), comprising 121,700 female nurses, and NHS II, consisting of 116,671 female nurses, and an effect modification of CS was found among smokers with > 10 pack-years in the NHS cohort, but not in the NHS II cohort. Thus, further studies in independent populations are necessary for validation.

#### 3.4.4. High Salt Intake

According to the case-control study (386 cases and 1886 matched controls) conducted by Sundström et al., sodium intake more than doubled the risk of RA among smokers, which was not observed among non-smokers. Moreover, the risk was further increased in the development of ACPA-positive and/or SE-positive RA cases, indicating a possible interactive effect between CS and high sodium intake on ACPA-positive RA.

## 4. Impacts of Cigarette Smoking on RA Pathogenesis

### 4.1. Effects of Cigarette Smoking on Immune Systems

CS affects both innate and adaptive immune responses, leading to altered cellular and humoral immunity to cause a systemic inflammation [74].

Skewed helper T (Th) cell subsets (Th1, Th2, and Th17) were observed depending on the relation to specific diseases [75,76,77], and Th1-skewness was reported among RA patients with CS [78]. In contrast, asthma, a Th2-skewed disease, has been reported to be associated with risk of RA [62,79,80,81,82,83,84,85,86], and CS is one of the well-established risks of asthma [87]. Activation of Th17 cells via aryl hydrocarbon receptor (AHR) [88] was also reported. Because Th-skewness is flexible and dynamic in the same individual with RA depending on various factors, such as medications, which, in turn, may influence disease activity or clinical course [89], it will be intriguing to further investigate how Th-skewness contributes to the pathogenesis of RA in relation to CS.

Increased levels of several pro-inflammatory cytokines (tumor necrosis factor (TNF)-α; interleukin (IL)-1α, IL-1β, IL-5, IL-6, IL-8, IL-13, IL-15, and IL-21; and interferon (IFN)-γ) in smokers with systemic autoimmune diseases including RA have also been well-documented [74,90,91,92,93,94,95]. Indeed, drugs targeting several of these cytokines, TNF-α, IL-6, and IL-1β, are currently used as biologic DMARDs for the treatment of RA [96] as well as JIA [97].

According to the study conducted by Glossop et al., CS increases TNF-α production from T cells, and both intensity and duration of CS are correlated with higher TNF-α/soluble TNF receptor (sTNFR) ratios in RA patients. Furthermore, smokers had higher ratios of TNF-α/sTNFR than non-smokers, suggesting that higher levels of TNF-α or ratios of TNF-α/sTNFR in smokers might be associated with TNF-α antagonist treatment resistance [94].

It was reported that serum levels of soluble IL-2 receptor (sIL-2R) were higher in smokers [98,99]. Furthermore, sIL-2R can affect the response to infliximab in RA patients, and a low serum sIL-2R level predicts rapid response to infliximab [100], suggesting that IL-2-sIL-2R activation may affect the response to anti-TNF-α treatment in RA patients with CS.

Elevated serum IFN-γ levels [95] and IFN-γ secretion both from effector CD4 and CD8 T cells have also been documented in RA patients [101]. As for the effects of CS, Bidkar et al. showed that CS exposure induced IFN-γ secretion from splenocytes of humanized transgenic (Tg)-mice carrying RA-susceptible *HLA-DRB1*0401*, while CS exposure augmented Th2 response in Tg-mice carrying RA-resistant *HLA-DRB1*04:02*. Despite the limitations of a mouse study, this implied a possible interaction of CS with the host *HLA* genes, leading to modulation of host immunity [102]. As mentioned above, CS can promote both Th1 and Th2 polarization. CD8 T cells, which are another major source of IFN-γ, have been reported to increase in number and be more prone to secreting cytokines due to CS in patients with chronic obstructive pulmonary disease (COPD) [103,104,105,106,107], and low soluble programmed death protein 1 ligand (sPD-L1) levels [108] and increased activated cytotoxic CD8 T cells [109] were also reported in RA patients. In contrast, the effects of CS on natural killer (NK) cells, which have similar cytotoxic functions as CD8 T cells, were variable in terms of the numbers and functions depending on an individual’s health status [110,111,112,113,114,115,116]. Thus, the effect of CS on IFN-γ production might be cell-type specific under the influence of the genetic background of an individual [117], which can eventually contribute to form a specific disease phenotype including RA.

Other mechanisms of CS suggested to affect immune systems with regard to RA development, such as autoimmunity to vimentin including induction of carbamylated vimentin [118], are also intriguing and are thus expected to be studied further.

### 4.2. Interactive Effects of Cigarette Smoking and Genetic Components

The most rheumatic diseases show complex traits with interactions between multiple genetic and environmental factors. Likewise, gene–environment interactions also play a critical role for RA pathogenesis (Figure 1).

#### 4.2.1. Interactive Effects between CS and the *HLA-DRB1* Gene on RA Development

CS has been implicated for its interactive effect, especially in relation to *HLA-DRB1* in seropositive RA cases. Several studies, not limited to Caucasian populations and including two Asian cohort studies, have investigated the interactive association between CS and ACPA formation in the context of *HLA-DRB1* alleles [119,120,121,122,123,124,125]. These studies, with the exception of a single North American cohort (SONORA), reported an interactive effect between CS and SE on ACPA-positive RA development. While most of the recent studies have focused on ACPA positivity, there have also been several studies that have investigated the interaction between CS and SE on RF-positive RA cases. Padyukov et al. observed that both CS and SE alleles conferred increased risk of RF-positive RA development, and there was a strong interaction between these two risks in the Swedish cohort (858 cases and 1048 controls) [126]. In contrast, the intra-case analysis conducted by Mattey et al. in 371 northern European white RA cases revealed that CS and *HLA-DRB1*04:01* were independently associated with RF production [127]. Thus far, the effect of the interaction between CS and SE on ACPA production is clear, while that on RF production is still debated—partly due to lack of recent evidence—and thus requires more research.

#### 4.2.2. Interactive Effects between CS and the *HLA-DRB1* Gene on RA-Related Autoantibody Production

While the interactive association of SE and CS with ACPA positivity is well documented, the effects on levels of ACPA as well as RF have not been studied well. In our recent study of Japanese RA cases, CS affected not only positivity but also levels of both ACPA and RF. The effect of CS was dependent on SE presence for ACPA but independent of SE status for RF [64]. Hedström et al. also reported similar findings in a case-control study with 3645 cases and 5883 controls. In the subset of patients positive for both RF and ACPA and the subjects only with positive ACPA, both CS and SE conferred independent risks, and there was a strong interaction between CS and SE. In the subset of patients with only positive RF, there was an increased risk of disease among smokers, which was only marginally affected by SE, and no interaction between CS and SE was observed. In the subset of patients negative for both RF and ACPA, neither CS nor SE conferred an increased RA risk [128]. These studies strongly indicate the different effect of CS on the development of ACPA and RF with regard to the interaction with SE alleles, highlighting the distinctive pathogeneses in different subsets of RA patients.

Furthermore, the study conducted by van der Helm-van Mil et al. suggested that interactive effects with CS were different among SE subsets; the interaction was strongest for the *HLA-DRB1*01:01* or **01:02* and *HLA-DRB1*10:01* SE alleles [124]. In contrast, Lundström et al. also reported that all SE alleles tested (**01*, **04:01*, **04:04*, **04:05*, **04:08*, and **10:01*) strongly interacted with CS in conferring an increased risk of ACPA-positive RA, regardless of the fine specificity of SE in the EIRA cohort (1319 cases and 943 controls) [129]. We also did not observe a difference between *HLA-DRB1*04:05* and non-**04:05* SE alleles in the interactive effect with CS on ACPA levels, and **09:01* did not show an interactive effect with CS on ACPA levels in Japanese RA cases [64]. In contrast, Bang et al. observed the different effect of interaction of *HLA-DRB1* alleles with CS on ACPA-positive RA development in a Korean case–control study (1924 cases and 1119 controls) [130]. Among the *HLA-DRB1* alleles they tested (five SE alleles, **01:01*, **04:01*, **04:04*, **04:05*, and **10:01*, and a non-SE allele, **09:01*, frequently observed in Asian populations), **10:01* showed the strongest interaction with CS, and the genotype heterozygous for **04:05* and **09:01* conferred the highest risk of both ACPA-positive and -negative RA development in the interaction with CS. These discrepant results may be partly due to the different frequencies of each SE allele among different ethnicities, and thus, meta-analyses or well-powered multi-ethnic studies are necessary to draw a solid conclusion.

#### 4.2.3. Interactive Effects between CS and the PTPN22 Gene on RA Pathology

Mahdi et al. reported specific interactive effects of CS and SE, or *PTPN22* (620W allele), one of the major GWAS genes and a potential causal variant in RA [131], on citrullinated α-enolase in a case-control study (1000 cases and 872 controls) [132]. The same group further extended the study by stratifying ACPA-positive RA patients into 17 subsets based on their profiles of different ACPA specificities (α-enolase, vimentin, fibrinogen, and collagen type II), which revealed the strongest association of SE, *PTPN22*, and CS in the subset of patients with antibodies to citrullinated α-enolase and vimentin [133]. In contrast, Willemze et al. showed that SE and CS promoted nonspecific citrullination rather than citrullination of specific antigens (α-enolase, vimentin, fibrinogen, and myelin basic protein) in Dutch RA patients with ACPA (661 cases) [134]. Fisher et al. also reported that the interaction between SE and CS was not exclusive to any of the specific citrullinated peptides (α-enolase, vimentin, and fibrinogen) in Korean RA patients (513 cases and 1101 controls) [135]. These studies strongly indicated the possible interactive effect of CS and SE on protein citrullination, with the specificity still remaining unknown, which in turn leads to ACPA formation.

#### 4.2.4. Interactive Effects between CS and the PADI4 Gene on RA Pathology

Kochi et al. found that *PADI4* polymorphism (rs1748033) predisposed male smokers to RA in a total of 2018 cases and 2035 controls from Japanese samples and also observed similar trends in a total of 635 cases and 391 controls from Dutch samples [136]. The study also showed that *PADI4* polymorphism, rs11203367, was significantly associated with ACPA status in ever-smokers in a recessive model, suggesting that *PADI4* polymorphism may be involved in the appearance of ACPA in smokers.

In summary, the gene-environment interaction, especially SE and CS, is strongly indicated in ACPA formation, while the effect on RF formation, if any, may be weaker than that on ACPA, although recent studies focusing on RF positivity are relatively lacking. As we mentioned in a previous section, seropositive polyarthritis JIA shares genetic components that confer a risk or protection with adult seropositive RA [25,26], and thus, it will be of interest to examine if the same effects seen in adult RA patients can also be found in the subset of JIA, especially with regard to passive smoking.

### 4.3. Effects of CS on Epigenetic Changes

CS causes wide-spread genome-scale changes in DNA methylation. In the epigenome-wide association study (EWAS) of the Swedish EIRA cohort (354 cases and 337 controls), Liu et al. identified two clusters of differentially methylated regions within the MHC region. By correcting cellular heterogeneity to adjust for cell-type proportions and with the use of analysis to filter out associations likely to be a consequence of disease, four CpGs also showed an association between genotype and variance of methylation, one of which was significantly associated with both clusters and the rest of which also showed suggestive association [76].

Zeilinger et al. conducted an independent EWAS with the use of large German populations, a total of 1814 for discovery and a total of 479 for replication, and found wide-spread differences in the degree of site-specific methylation as a function of CS in each of the 22 autosomes, confirming the broad effect of CS on epigenetic changes. Among the observed changes, methylation-specific protein binding patterns observed for cg05575921 within aryl hydrocarbon receptor (*AHR*) repressor (*AHRR*) had the highest level of changes in DNA methylation associated with CS, suggesting a regulatory role for gene expression. Importantly, methylation levels in past-smokers were close to the ones seen in never-smokers depending on cessation time and pack-years [137].

As for the interactive effects of CS with gene polymorphisms on RA epigenome, Meng et al. identified a significant interaction between rs6933349 of mucin *22* (*MUC22*) and CS in DNA methylation of cg21325723 in terms of the risk of developing ACPA-positive RA in both Caucasian and Asian populations [138].

Although data for epigenetic phenomena in RA are currently limited in terms of study scale and power [139], it is highly likely that epigenetic changes play crucial roles in the development of RA via gene regulation. Further study will be needed with considerations of sample throughput methods and genome coverage and resolution, such as the use of whole-genome bisulfite sequencing [140]. Longitudinal cohorts will also be essential for establishing the temporal origin of deleterious events and distinguishing causal from consequential effects [141].

### 4.4. Cigarette Smoking Modulates Periodontal Disease Leading to a Higher Risk of RA Development

It is well established that CS is one of the risks of periodontal disease (PD) [142], and there have been many studies that have linked CS-affected PD development with a higher risk of RA development. As we mentioned in the previous section, *P. gingivalis*, a major periodontal pathogen [143,144], can induce ACPA by bacterial PAD, and worsen the severity of RA [145,146,147]. Another major pathogen, *A. actinomycetemcomitans* is also known to be associated with hypercitrullination in the gingival tissues [148]. Furthermore, periodontitis is known to correlate with ACPA levels in healthy individuals [149] and often precedes RA development [150,151]. Notably, RA and periodontitis also shared genetic risks, such as SE alleles [152]. Not only ACPA but also RF positivity is associated with periodontitis [142,153], although the underlying mechanisms might be different between these autoantibodies [64,128]. Collectively, CS can also be a strong risk modifier of RA development via its effect on PD pathology, especially in individuals with SE alleles.

### 4.5. Airway Inflammation

CS significantly increases the number of alveolar macrophages and other monocytes, which, in turn, increases levels of lysosomal enzymes and secrete elastase responsible for parenchymal and connective tissue damage [74]. Demoruelle et al. demonstrated that airway inflammation is common in healthy ACPA-positive subjects before clinically apparent RA development [154]. Klareskog et al. also demonstrated that CS induces protein citrullination in the lungs [119]. Thus, one interesting possible hypothesis is that initial inflammation and immune abnormality of RA may generate from the lungs [154].

Matrix metalloproteinases (MMP)-12 has been implicated in the pathogenesis of RA [155], and animal experiments have suggested MMP-12 as one potential mediator of airway inflammation. For example, MMP-12 expression was increased in macrophages and dendritic cells in the lungs of CS-exposed mice [156]. Other MMPs, proMMP-2 and proMMP-9, have also been reported to be increased in the sera of smokers [157]. Although RA synovial fibroblasts-derived MMP-9 may directly contribute to joint destruction in RA [158], further investigation as to whether the effects of these MMPs on RA pathogenesis are dependent on airway inflammation or not is required.

### 4.6. The Role of Passive Smoking in RA Pathogenesis and the Role of Nicotine

Although nicotine is a main toxic substance in both direct smoking and passive smoking, passive smoking in childhood may have distinctive mechanistic impacts on RA pathogenesis from direct CS considering that the smoke from the burning end of a cigarette has more toxins than the smoke inhaled by smokers, and passive smoking in childhood increases the risk of asthma [159], which may be associated with RA risk [62].

Interestingly, Jian et al. reported that the risk of CS on both ACPA-positive and -negative RA development increased only with the use of inhaling cigarettes but not nicotine-contained chewed cigarettes (OR, 1.0; 95% CI, 0.8–1.2), suggesting that nicotine is not directly involved in RA pathogenesis [160]. Indeed, nicotine inhibits TNF-α-induced IL-6 and IL-8 secretion in fibroblast-like synoviocytes (FLS) from RA patients [161]. Moreover, several animal experiments showed an immunosuppressive effect of nicotine, such as the loss of antibody response and T-cell proliferation [162,163,164]. In contrast, the paradoxical effect of nicotine on RA has also been reported; nicotine pretreatment aggravated adjuvant-induced arthritis (AIA), whereas post-treatment with nicotine suppressed the disease in rats [165]. Furthermore, blood cotinine, a nicotine metabolite, was positively correlated with the prevalence of naive CD3+ T cells among non-smokers exposed to passive smoking [166]. Thus far, the role of nicotine in the pathogenesis of RA is still questionable and needs further investigation with regard to the effect of passive smoking on RA.

## 5. Concluding Remarks and Future Directions of Studies

Numerous efforts have been made to clarify the pathogenesis of RA in relation to CS. Despite such efforts, the underlying mechanisms still have not been clarified due to the complex nature of the effects of CS on RA pathology. However, a growing body of data has suggested that environment–environment or gene-environment interactions are key mechanisms to trigger the onset and modify the course of the disease, and thus, future studies should take into consideration these interactions in association studies of CS and RA. In addition, most studies have found that the effect of CS on RA pathogenesis is limited or much stronger in seropositive RA, and effects of CS seem to be different between ACPA- and RF-positive subsets, which will also be an important subject of future studies. The different effect of CS among different ethnicities will also be of interest.

In summary, the effect of CS on RA pathology has multiple aspects; interaction with genetic components as well as other environmental factors, effects on immune systems including both innate and acquired immunity, and epigenetic changes by several key chemical compounds or reactive oxygen species (ROS), which altogether might contribute to pathogenesis. Well-powered studies with consideration of several key points mentioned above will further clarify the precise pathogenic role of CS, which will lead to our better understanding of RA pathogenesis and the development of better treatment options.

## Figures and Tables

**Figure 1 cells-09-00475-f001:**
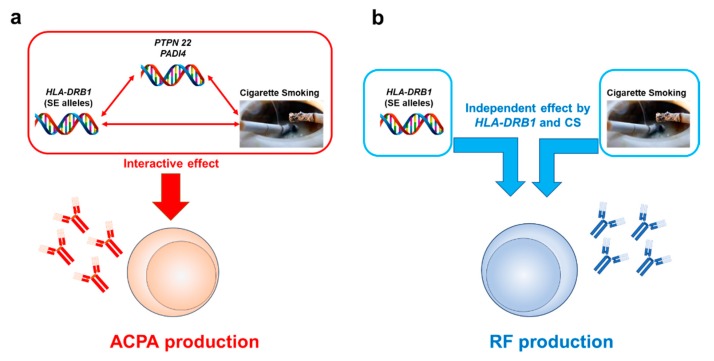
The differential effects of gene and environmental factors (cigarette smoking) on ACPA and RF production. The interactive effects of cigarette smoking (CS) and genetic components, especially *HLA-DRB1*, contribute to ACPA production and subsequently predispose subjects to ACPA-positive RA development. *PTPN22* and *PADI4* are also implicated as additional genetic components (**a**). For RF production, an interactive effect between genetic and CS is much less clear; rather, they are independent risk factors (**b**).

**Table 1 cells-09-00475-t001:** The risks of cigarette smoking for rheumatoid arthritis (RA) development, RA-related pathologies, and comorbidities.

First Authors	Study Type	Outcomes	Effects and Effect Sizes	Interaction between CS and Genetic Components	Stratifications	Population, Country, Study Period
Di Giuseppe	Meta-analysis	RF (+) or (−) RA development	Dose-dependent increase of RR (1.26–2.07) up to 40 pack-years; RR 2.47 and 1.58 for RF (+) and (−) RA, respectively	NA	Pack-years; RF	Three cohorts and seven case-control studies; a total of 4552 RA cases
Hedström	Case-control	ACPA (+) or (−) RA development	OR 1.9 and 1.3 for ACPA (+) and (−) RA, respectively; a dose-response association (p for trend < 0.0001); cessation > 20 years diminishes the risk of ACPA (−) RA	NA	Never-, ever-, past, current smokers; duration; intensity; pack-years; ACPA	3655 cases and 5883 matched controls in Sweden
Hedström	Case-control	ACPA (+) or (−) RA development	No association between passive smoking and RA risk (OR ~ 1.0 for both ACPA (+) and (−) RA)	NA	Duration of exposure; ACPA	589 cases and 1764 controls without smoking history
Seror	cohort	RA development	Only a suggestive risk of passive smoking (HR1.4–1.7)	NA	Never- or ever-smokers w/or w/o passive CS during childhood	71,248 French female volunteers prospectively followed since 1990; 371 RA cases
Kim	intra-case	Clinical response	Better clinical response in never-smokers than in passive smokers	NA	Never, current, ex-, and passive smokers	191 female RA cases in South Korea
Torrente-Segarra	intra-case	Clinical response	Better clinical response in never- than in passive smokers, which does not result in better drug survival	NA	Smoking status, ACPA	1349 RA cases from METEOR database between 2006 and 2016
Rydell	intra-case	Radiographic progression	OR 3.17 for RRP in ever-smokers	NA	Never-, current, ever-, and previous smokers	233 early RA cases during 1995–2005 in Sweden
Sivas	intra-case	Disease activity, radiographic score	Higher erosion and joint space narrowing scores in smokers; no correlation of smoking with disease activity	NA	Never-, long-term, and new smokers	165 Turkish RA cases (129 females) followed between January 2015 and February 2016
van Wesemael	Case-control	RF, ACPA, and anti-CarP Ab presence	Smoking was associated with multiple autoantibody positivity both in non-RA and RA cases (OR 1.32–2.95)	NA	Never- and ever-smokers; ACPA, RF, anit-CarP Ab	9575 Japanese non-RA subjects; early RA cases from the Netherlands (*n* = 678), UK (*n* = 761), and Sweden (*n* = 795)
Ishikawa	intra-case	RF or ACPA positivities and levels	OR of CS 2.06 and 1.29 for high levels of RF and ACPA, respectively	Interactive effect of CS and SE on ACPA levels but not those of RF	Never-smokers, ex- or active smokers at the onset; SE; ACPA; RF	6239 Japanese RA cases
Klareskog	Case-control	ACPA (+) or (−) RA development	Dose-dependent effect of CS on ACPA (+) RA development	Interactive effect between CS and SE on ACPA (+) RA	Never and ever-smokers; pack-years; numbers of SE; RF; ACPA	913 early RA cases and 1357 controls, Sweden
Too	Case-control	ACPA (+) or (−) RA development	OR of CS 4.1 and OR of SE 4.7 for ACPA (+) RA development	Interactive effect between CS and SE on ACPA (+) RA	Never- and ever-smokers; SE; ACPA; RF	1076 early RA cases and 1612 matched controls, Malaysia, 2005–2009
Lee	intra-case	ACPA (+) or (−) RA development	Correlation between CS and ACPA (+) RA was observed in 2 out 3 cohorts.	Weak interaction between CS and SE for ACPA only in one cohort	Never- and ever-smokers; SE; ACPA; RF	A total of 2476 white patients with RA from three different cohorts, North America
Bang	Case-control	ACPA or RF (+) or (−) RA development	OR of ever-smoking 2.22 for ACPA (+) and 2.80 for ACPA (−) RA	Interactive effect of CS and SE both on ACPA (+) and ACPA (−) subsets	Never- and ever-smokers; SE; DRB1*09:01; ACPA; RF	1482 RA cases and 1119 control subjects, Korea
Murphy	intra-case	ACPA or RF (+) or (−) RA development	Strong association between ACPA and RF but not ACPA and CS; no association of CS and ACPA in RF (−) cases	No interaction between CS and SE	Never- and ever-smokers; Pack-years; SE; ACPA; RF	Two different UK RA cohorts (*n* = 658 and 409)
van der Helm-van Mil	cohort	ACPA (+) or (−) RA development	*HLA–DRB1*0401*, **0404*, **0405*, or **0408* SE alleles conferred the highest risk of ACPA development	Strongest interaction between CS and *01:01 or *01:02 and *10:01 alleles	Current and past smokers; SE and subsets; ACPA	977 undifferentiated arthritis cases, Netherland
Pedersen	Case-control	ACPA (+) or (−) RA development	No significant effect of CS on SE (−) subjects	Strong interaction between CS and SE for ACPA (+) but not ACPA (−) RA	SE; ACPA; never- and ever-smokers; pack-years; coffee or alcohol consumption; oral contraceptive use	445 RA cases and 533 age- and sex-matched controls, Denmark, 2002–2004
Padyukov	Case-control	RF (+) or (−) RA development	Neither CS nor SE genes nor the combination increased the risk of RF (−) RA development	Significant interaction between CS and any SE genes on RF (+) RA	Gender, smoking status, and HLA-DRB1 genotypes, RF	RA 858 cases and 1048 controls recruited during 1996 to 2001, Sweden
Mattey	intra-case	RF (+) or (−) RA development	OR of ever-smoker for RF (+) RA development 2.2 in ever-smokers	independent effects of CS and SE, *HLA-DRB1*04:01*, on RF (+) RA	Never-, ever-, current smokers; SE and subsets; RF	371 RA cases, UK
Hedström	Case-control	ACPA or RF (+) or (−) RA development	An independent effect of CS on RF (+) but not on RF (−) RA regardless of ACPA status	Significant interaction between CS and SE on ACPA (+) RA	Never-, ever-, current smokers; SE; ACPA, RF	3645 cases, 5883 matched controls, Sweden; follow-up on Ref. 17
Lundström	Case-control	ACPA (+) or (−) RA development	Lack of an independent effect of CS on ACPA (+) RA	Significant interaction of CS with all SE genes tested on ACPA (+) RA	Never- or ever-smokers; SE (DRB1*04, *01, and *10); ACPA	RA 1319 cases and 943 controls recruited during 1996 to 2005, Sweden; partially overlapped with Ref. 119
Bang	Case-control	ACPA (+) or (−) RA; ACPA levels	Smokers had a trend of higher ACPA levels than never-smokers without significant difference	Significant interaction of CS with SE but not with **09:01* on ACPA (+) RA	Never- or ever-smokers; SE; DRB1*09:01; ACPA	1924 RA cases and 1119 control subjects, Korea; partially overlapped with Ref. 115
Mahdi	Intra case and case–control	Anti-CEP-1 Ab response	43–63% of ACPA (+) cases were anti-CEP-1 Ab (+), and this subset was preferentially linked to *HLA-DRB1*04.*	Combined effect of CS, PTPN22, and SE on anti-CEP (+) response	Never- or ever-smokers; SE; PTPN22; ACPA; anti-CEP	1497 cases, Sweden and UK; 1000 cases and 872 controls, Sweden (cases were overlapped)
Lundberg	Case-control	Specific ACPA responses	The strongest association of SE, PTPN22, and CS for the RA subset anti-CEP-1 (+) or anti-cVim Ab (+) subsets of RA		Never-, past, and current smokers; SE; PTPN22; ACPA subsets	1985 cases and 2252 matched controls, Sweden overlapped with Refs. 17, 121
Willemze	intra-case	Specific ACPA responses	A significant interaction between CS and SE for the presence of ACPA, not restricted to specific citrullinated antigens		Never- and ever-smokers; SE; ACPA subsets; RF; ANA	661 cases with recent onset (< 2 years), Netherland
Fisher	Case-control	Specific ACPA responses, erosion	CS-SE interaction was associated with all the ACPA (+) subgroups; highest OR in an anti-CCP (+) cVim (+) subset		Never- and ever-smokers; SE and DRB1*09:01; ACPA subsets	513 cases and 1101 controls, Korea overlapped with Ref. 115
Kochi	Case-control	RA development	PADI4 SNP (rs1748033) conferred a higher risk in men (OR 1.39) and in ever-smokers (OR 1.25)	The highest risk in male ever-smokers (OR 1.46)	Never- and ever-smokers; PADI4 SNP genotypes; gender; ACPA	1019 cases/907 controls and 999 cases/1128 controls, Japan; 635 cases/391 controls, Netherland

RA, rheumatoid arthritis; CS, cigarette smoking; RF, rheumatoid factor; RR, relative risk; NA, not assessed; ACPA, anti-citrullinated cyclic peptide/protein antibody; OR, odds ratio; HR, hazard ratio; RRP, rapid radiographic progression: anti-CarP Ab, anti-carbamylated protein antibody; SE, shared epitope; anti-CEP-1, anti-citrullinated α-enolase protein 1; anti-cVim, anti-citrullinated vimentin; SNP, single nucleotide polymorphism.

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
