# Peer review of "The Impact of Cigarette Smoking on Risk of Rheumatoid Arthritis: A Narrative Review"

_cells, 2020, doi:10.3390/cells9020475_

Round 1

Reviewer 1 Report

The review addresses an interesting topic; however, the paper is not well organized and needs an extensive revision. In the  section 2 (environmental risks of RA) there is a short paragraph on other enviromental risk that should be deleted since it is approached too superficially. The sections should be re-ordered starting from the genetic risk, the addition on environmental factors (cigarette smoking and other factors influenced/enhanced by smoking) than epidemiolgical data and finally clinical data.

Author Response

First of all, we deeply appreciated the reviewer’s comments and suggestions, all of which were helpful to brush up this manuscript.

The review addresses an interesting topic; however, the paper is not well organized and needs an extensive revision.

We extensively edited the original manuscript entirely to make this revision better organized and more schematic.

In the section 2 (environmental risks of RA) there is a short paragraph on other enviromental risk that should be deleted since it is approached too superficially.

We removed this section according to the both reviewers’ suggestion. Other environmental factors influenced by CS were left with clear description of their relation to CS.

The sections should be re-ordered starting from the genetic risk, the addition on environmental factors (cigarette smoking and other factors influenced/enhanced by smoking) than epidemiolgical data and finally clinical data.

In this revision, original section 4 and 5 were merged and sub-sections were created for a better organization. As the reviewer suggested, we also re-ordered the sections.

Reviewer 2 Report

The authors submitted an interesting (narrative) review discussing the evidences about the impact of cigarette smoking on the risk and pathogenesis of RA. In this regard, I would suggest the authors to declare the type of review (narrative) and the research parameters and methods they used to get the needed references.

- The introduction is informative enough.

- The section 2 (Environmental risks [ RISK FACTORS…could be better] of RA) is appropriate, as regards the first 4 paragraph, basically until the table. Actually, I would recommend deleting the second part, from “other instances” until the end; thus, as a consequence, I suggest renaming this section, in order to focus on CS only, which is the topic of this review. Of course, also Table 2 should be deleted, in my opinion. 

- Indeed, the authors would do better to develop the discussion about the referenced articles included in Table 1, by providing more details and information on the study context design, which cannot be completely included in the nice overview represented by Table 1.

- In detail, I would also suggest expanding the discussion by dedicating one or two specific paragraph(s) to the passive smoking in childhood both here and in the pathogenesis section later. Importantly, the authors may even consider the impact on juvenile idiopathic arthritis, at least as regards those rarer forms resembling RA, like RF-positive JIA, which actually shares some genetic similarities with adult RA as regards the HLA-background as well (refer to: HLA-DRB1 alleles and juvenile idiopathic arthritis: Diagnostic clues emerging from a meta-analysis. Autoimmun Rev. 2017 Dec;16(12):1230-1236. doi: 10.1016/j.autrev.2017.10.007; & Review: Genetics and the Classification of Arthritis in Adults and Children. Arthritis Rheumatol. 2018 Jan;70(1):7-17. doi: 10.1002/art.40350).

- Knowing the literature search criteria would very important to justify the selected article for Table 1. Indeed, a meta-analysis (Di Giuseppe et al., no country) is placed along with other types of study. Actually, this information (study type) should be highlighted directly in the table, in my opinion.

- Section 3: the discussion about the genetic risk of RA should focus more on the relationship with smoke exposure. By the way, both here and in the previous section, I think it is important to discuss separately active and passive smoking, for the sake of clarity.

- Section 4: an introduction (one or two paragraphs) about the immunologic pathogenesis of RA would be very useful to the readership and would help to better understand the impact of CS on the pathogenesis of RA in terms of effect that CS may have on key cytokines promoting RA. As mentioned by the authors previously and supported by myself as well, I think that highlighting the role of passive smoke exposure in childhood (separately and also in terms of pathogenesis) could add a significant value to this review.

- In this regard, the authors should refer to all those studies investigating the impact of CS on the level/production of those cytokines that are more implicated in RA/JIA pathogenesis, like IFN-gamma, TNF-alpha and others. As for IFN-gamma, additional references could be useful (e.g. Clin Rheumatol. 2019 Nov;38(11):3061-3071. doi: 10.1007/s10067-019-04681-4. 2016 Sep 7;11(9):e0162341. doi: 10.1371/journal.pone.0162341; etc.). A similar approach may be used for other key cytokines.

- In general, I would recommend adopting a more schematic approach all over the manuscript. Some more specific observations could be provided after the authors revise the manuscript according to all these general observations and recommendations.

- The conclusion should provide clearer take-home messages.

- The references should be completed and expanded according to the precise literature search criteria (to be declared) and the previous recommendations.

Author Response

First of all, we deeply appreciated the reviewer’s comments and suggestions, all of which were helpful to brush up this manuscript.

The authors submitted an interesting (narrative) review discussing the evidences about the impact of cigarette smoking on the risk and pathogenesis of RA. In this regard, I would suggest the authors to declare the type of review (narrative) and the research parameters and methods they used to get the needed references.

We clearly described the information about the type of review in the title. We also add the parameters we adopted for literature search in the end of Introduction.

- The introduction is informative enough.

We appreciate the reviewer’s supportive comment.

- The section 2 (Environmental risks [ RISK FACTORS…could be better] of RA) is appropriate, as regards the first 4 paragraph, basically until the table. Actually, I would recommend deleting the second part, from “other instances” until the end; thus, as a consequence, I suggest renaming this section, in order to focus on CS only, which is the topic of this review. Of course, also Table 2 should be deleted, in my opinion. 

We removed this section as well as Table2 according to the both reviewers’ suggestion. Other environmental factors influenced by CS were left with clear description of their relation to CS.

- Indeed, the authors would do better to develop the discussion about the referenced articles included in Table 1, by providing more details and information on the study context design, which cannot be completely included in the nice overview represented by Table 1.

We updated Table 1 by adding further information including types of study.

- In detail, I would also suggest expanding the discussion by dedicating one or two specific paragraph(s) to the passive smoking in childhood both here and in the pathogenesis section later. Importantly, the authors may even consider the impact on juvenile idiopathic arthritis, at least as regards those rarer forms resembling RA, like RF-positive JIA, which actually shares some genetic similarities with adult RA as regards the HLA-background as well (refer to: HLA-DRB1 alleles and juvenile idiopathic arthritis: Diagnostic clues emerging from a meta-analysis. Autoimmun Rev. 2017 Dec;16(12):1230-1236. doi: 10.1016/j.autrev.2017.10.007; & Review: Genetics and the Classification of Arthritis in Adults and Children. Arthritis Rheumatol. 2018 Jan;70(1):7-17. doi: 10.1002/art.40350).

We added one paragraph describing the impact of passive smoking on risk of RA in this section. We also add a description of passive smoking in terms of its potential mechanistic impacts on RA pathogenesis in Section 4. We also briefly touched genetic similarities of seropositive polyarthritis JIA with adulthood RA both in Section 2 and 4 referring the suggested papers.

- Knowing the literature search criteria would very important to justify the selected article for Table 1. Indeed, a meta-analysis (Di Giuseppe et al., no country) is placed along with other types of study. Actually, this information (study type) should be highlighted directly in the table, in my opinion.

We described literature-search criteria in Introduction. We also updated Table 1 by adding further information including types of study.

- Section 3: the discussion about the genetic risk of RA should focus more on the relationship with smoke exposure. By the way, both here and in the previous section, I think it is important to discuss separately active and passive smoking, for the sake of clarity.

We first discussed genetic risks of RA comprehensively without focusing on any other risks in revised section 2, and then, as the reviewer suggested, we further discussed the genetic risks focusing on their relation to CS in section 4 of the revision.

According to the reviewer’s suggestion, we discussed active and passive smoking separately both in section 3 and 4 of this revision, trying to suggest their potentially different effects on RA pathogenesis from that of active smoking.

- Section 4: an introduction (one or two paragraphs) about the immunologic pathogenesis of RA would be very useful to the readership and would help to better understand the impact of CS on the pathogenesis of RA in terms of effect that CS may have on key cytokines promoting RA. As mentioned by the authors previously and supported by myself as well, I think that highlighting the role of passive smoke exposure in childhood (separately and also in terms of pathogenesis) could add a significant value to this review.

- In this regard, the authors should refer to all those studies investigating the impact of CS on the level/production of those cytokines that are more implicated in RA/JIA pathogenesis, like IFN-gamma, TNF-alpha and others. As for IFN-gamma, additional references could be useful (e.g. Clin Rheumatol. 2019 Nov;38(11):3061-3071. doi: 10.1007/s10067-019-04681-4. 2016 Sep 7;11(9):e0162341. doi: 10.1371/journal.pone.0162341; etc.). A similar approach may be used for other key cytokines.

We added immunologic pathogenesis of RA in section 4, in which the effects of CS on altered skewness of helper T cell subsets and cytokine secretion from immune cells were discussed. We also mentioned passive smoking here again to suggest a potentially different effects on RA pathogenesis from that of active smoking.

- In general, I would recommend adopting a more schematic approach all over the manuscript. Some more specific observations could be provided after the authors revise the manuscript according to all these general observations and recommendations.

We extensively edited the original manuscript entirely to make this revision better organized and more schematic.

- The conclusion should provide clearer take-home messages.

We also reorganized this section to provide readers with important key messages more clearly.

- The references should be completed and expanded according to the precise literature search criteria (to be declared) and the previous recommendations.

We described literature search criteria in the last part of Introduction in this revision. We also expanded literature search in line with this revision according to the search criteria described above.

Round 2

Reviewer 1 Report

the autors exentsively modified the manuscript according to the issue raised-

I suggest to accept the review.

Author Response

Thanks!

Reviewer 2 Report

The authors addressed appropriately most of my previous observations and comments. However, some minor corrections are still needed and few specific points should discussed more and/or highlighted, as explained below.

TITLE: keep “risk” rather than “risks”. ABSTRACT: avoid too many abbreviations in the abstract (CS) I would like to ask for both tracked and clean copies of the revised manuscript for the next re-review. Indeed, it was difficult to clearly understand the final corrections through this PDF format, especially if they are many, involve long part of the text and close each other. See introduction. Anyway, I guess the old section 2. Environmental risks of RA has been completely deleted, according to the previous comments. the current section 2. “Genetic risks of RA” and related subsections are appropriate. One minor comment about the title. I suggest “genetic risk factors of RA” or “genetic risk of RA”. Moreover, abbreviations in the titles should be avoided. The references are appropriate in general. Section 3 makes much more sense now. Indeed, the authors significantly improved the general organization of the discussion. No major comments. Sections 4 improved as well. The authors tried to summarize the potential contribution of several specific immunological factors/cytokines. Some more or recent paper focusing on human studies may be retrieved. As for IFN-gamma, the authors highlighted itspotential role, which is proposed by several researchers (e.g. “Are key cytokines genetic and serum levels variations related to rheumatoid arthritis clinical severity? Gene. 2020 Jan 5;722:144098. doi: 10.1016/j.gene.2019.144098. However, most studies in the literature report an impaired production of IFN-gamma in association with both passive and active cigarette smoking. This apparent mismatch/conflict should be reported or deserves to be highlighted by the authors by adding some sentences to this short paragraph, considering the main subject (CS and its interactions with several genetic and environmental factors) The rest of section 4 is clear enough, but there some typing mistakes (see “4.3 Effects of CS on eipgenetic changes”) the second part of the concluding remarks should be revised and provide clear take home messages. Again, a clean copy along with a tracked copy, would be very helpful to read and revise the manuscript.

Author Response

The authors addressed appropriately most of my previous observations and comments. However, some minor corrections are still needed and few specific points should discussed more and/or highlighted, as explained below.

First of all, we appreciate all the comments by this reviewer, which were very supportive and helpful to make this manuscript improved.

TITLE: keep “risk” rather than “risks”.

We amended the title as the reviewer suggested.

ABSTRACT: avoid too many abbreviations in the abstract (CS)

We avoid the abbreviation, “CS”, in the abstract of this revision.

I would like to ask for both tracked and clean copies of the revised manuscript for the next re-review. Indeed, it was difficult to clearly understand the final corrections through this PDF format, especially if they are many, involve long part of the text and close each other. See introduction.

We are sorry for your inconvenience. Indeed, we had sent a tracked copy of the previous revision as well as a clear copy, but probably it was not offered to you from the editorial office. We again send both clean and tracked copies of this revision so that you can refer the changes we have made in this revision.

Anyway, I guess the old section 2. Environmental risks of RA has been completely deleted, according to the previous comments. the current section 2. “Genetic risks of RA” and related subsections are appropriate. One minor comment about the title. I suggest “genetic risk factors of RA” or “genetic risk of RA”. Moreover, abbreviations in the titles should be avoided.

We amended the title as the reviewer suggested.

The references are appropriate in general.

We appreciate the reviewer’s supportive comments.

Section 3 makes much more sense now. Indeed, the authors significantly improved the general organization of the discussion. No major comments.

We appreciate the reviewer’s supportive comments.

Sections 4 improved as well. The authors tried to summarize the potential contribution of several specific immunological factors/cytokines. Some more or recent paper focusing on human studies may be retrieved. As for IFN-gamma, the authors highlighted its potential role, which is proposed by several researchers (e.g. “Are key cytokines genetic and serum levels variations related to rheumatoid arthritis clinical severity? Gene. 2020 Jan 5;722:144098. doi: 10.1016/j.gene.2019.144098. However, most studies in the literature report an impaired production of IFN-gamma in association with both passive and active cigarette smoking. This apparent mismatch/conflict should be reported or deserves to be highlighted by the authors by adding some sentences to this short paragraph, considering the main subject (CS and its interactions with several genetic and environmental factors)

We appreciate the reviewer’s helpful comments. We briefly described conflicting reports of the effect of cigarette smoking on IFN-γ production referring several human studies in the corresponding paragraph.

The rest of section 4 is clear enough, but there some typing mistakes (see “4.3 Effects of CS on eipgenetic changes”) the second part of the concluding remarks should be revised and provide clear take home messages.

We are really sorry for the typo. We went through the whole manuscript again to correct any mistakes. We also added clear take-home messages in the concluding remarks.

Again, a clean copy along with a tracked copy, would be very helpful to read and revise the manuscript.

As we mentioned above, we send both clean and tracked copies of this revision so that you can refer the changes we have made in this revision.

Round 3

Reviewer 2 Report

No additional comments.